# The Safety of Topical and Intravenous Tranexamic Acid in Endoscopic Sinus Surgery

Agrani Ratnayake Kumar [1,2] and Andrew James Wood [1,2,3,*]

1    Waikato Clinical Campus, The University of Auckland, Hamilton 3204, New Zealand
2    Waikato Institute of Surgical Education and Research, Hamilton 3204, New Zealand
3    Department of Otolaryngology, Waikato Hospital, Hamilton 3204, New Zealand
*    Correspondence: andrew.wood@auckland.ac.nz

**Abstract:** Tranexamic acid (TXA) is an inexpensive and widely used medication indicated for the reduction of bleeding. There are data showing the efficacy of intravenous (IV) and topical TXA in endoscopic sinus surgery (ESS) but the safety of this practice is not well studied. The objective of this study was to assess the safety of using both IV and topical TXA in ESS. A scoping review was performed to investigate the effect of TXA on respiratory epithelia. A retrospective single-surgeon study was used to assess 177 comprehensive ESS cases from January 2017–December 2019 for the safety of combined IV and topical TXA. The scoping review demonstrated that respiratory epithelia could withstand a wide range of TXA concentrations without detrimental morphological effects. Topical TXA may have positive effects on wound healing and inflammation. The retrospective study showed no thromboembolic complications attributable to TXA in the 28 days after ESS. Only two patients (1.3%) who received TXA re-presented with post-operative bleeding. The use of IV and topical TXA is safe with regards to its effect on respiratory epithelium and thromboembolic disease. Topical TXA may have more positive effects than merely the reduction of bleeding following ESS.

**Keywords:** tranexamic acid; thromboembolism; respiratory mucosa; wound healing; surgical procedures





## 1. Introduction

Bleeding during and after endoscopic sinus surgery (ESS) is common and causes management dilemmas. A UK audit from 1998–2002 found that 58% of all re-admissions within 24 h of ESS were due to post-operative bleeding [1]. Similarly, a more recent New Zealand study found that post-operative bleeding accounted for 80% of the total complications after day-case sinonasal surgery, with 6 out of 117 patients re-presenting owing to bleeding [2]. Increased bleeding during surgery results in the poor display of structures and anatomical landmarks, which can further increase the rate of complications [3,4].

The ideal nasal dressing after ESS is one that is hemostatic but also improves mucosal healing [5]. Tranexamic acid (TXA) is a readily available and inexpensive medication that has the potential to contribute as such a nasal dressing. TXA has an anti-fibrinolytic action, blocking the lysine-binding sites of plasminogen. This prevents the activation of plasmin and subsequent degradation of fibrin products. The secondary anti-fibrinolytic effect is to block the lysine-binding sites of free plasmin that has already formed. Overall, this maintains the fibrin clot and reduces bleeding.

The efficacy of intravenous (IV) TXA in ESS is largely proven, with a recent systematic review and meta-analysis showing significantly lower operative time and higher surgeon satisfaction when used intra-operatively [6]. There were also no significant differences in experimental groups with regards to post-operative nausea, vomiting, coagulation profiles, and thrombotic events [6]. Topical administration of TXA is also attractive as it maximizes the drug effect at the local site whilst minimizing the side effects induced by systemic drug exposure. Although there have been fewer studies addressing the use of topical TXA in

ESS, the available studies imply improved surgeon satisfaction and subjective surgical field quality, and reduced blood loss and surgical time [7–10].

In the nasal vestibule, the epithelium transitions from non-keratinized stratified squamous epithelium (skin) to pseudostratified ciliated (respiratory) epithelium [11]. The respiratory epithelium covers the sinuses, conchae, meatuses, septum, and the floor of the nasal cavity [11]. Therefore, these are the cells most exposed to TXA when used topically in ESS. However, there is limited evidence assessing the safety of direct application of TXA in the sinuses and the nasal cavity, specifically with regards the respiratory epithelium.

The safety of IV TXA with regards to myocardial infarction (MI), stroke, deep vein thrombosis (DVT), and pulmonary embolism (PE) is also still uncertain [12]. Studies that assessed the use of either IV or topical TXA in ESS had a maximum follow-up period of 3 days, which was inadequate to sufficiently assess post-operative adverse events [7–10]. It has been shown that following gynecologic cancer surgery, 76% of thromboembolic events occurred after post-operative day seven [13]. In a study of 5607 lower limb joint procedures, the mean time to diagnosis of DVT was 21 days after total hip replacement and 20 days after total knee replacement [14]. More recently, in a large multi-national study of non-cancer general-surgical patients, 77% of the thromboembolic events were detected after the first post-operative week [15].

Currently, the practice of the senior author of this paper is to request administration of IV TXA at induction of anesthesia and to apply topical tranexamic acid (5 mL of 100 mg/mL) via cotton pledgets placed within the sinus cavities at the conclusion of surgery. The pledgets are then removed in the post-anaesthetic recovery room 60 min later. No other nasal or sinus packing material is used. Given the relative abundance of data regarding the efficacy of TXA in ESS but the paucity of appropriate safety data, the primary objective of this study was to evaluate the safety of TXA in ESS, both with regards to local effects on the sinonasal mucosa and thromboembolic complications.

## 2. Materials and Methods

There were two components to this study. First, a scoping review was used to assess the effects of TXA when applied directly to respiratory epithelia. Second, a retrospective clinical study was used to analyze the safety of using both IV and topical TXA in ESS.

### 2.1. Scoping Review

2.1.1. Databases Used

Multiple databases were searched by the first author, including Medline (Ovid) (1946–present), Embase (1947–2020), Scopus, Biosis (1945–2020), ProQuest, and Web of Science, in March 2020 and again in November 2020. Grey literature was sought on OpenGrey and E-thesis Online Service (EThOS). The bibliography list of all the included studies were hand-searched against the inclusion and exclusion criteria. The following search terms were used: "Tranexamic acid" OR "Anti fibrinolytic" OR "AMCA" OR "t-AMCA" OR "Cyklokapron" AND "Nasal epithelium" OR "Respiratory epithelium" OR "nasal mucosa". OR "Respiratory mucosa".

2.1.2. Eligibility Screening Criteria

Studies that used respiratory epithelia derived from either animal or human cells that were exposed to any form of TXA by any method of administration were included in the review. In vivo, in vitro, ex vivo, or cadaveric experimental conditions were accepted. Reviews and articles where only an abstract was presented, and articles that were not written in English and did not have an available translation were excluded. Studies which focused exclusively on other tissue types, such as non-respiratory epithelia or chondrocytes, were also excluded from this review.

*2.2. Retrospective Clinical Study*

A single surgeon retrospective study of all cases of comprehensive ESS at Waikato Hospital between January 2017 and December 2019 was performed. A logbook of all surgical cases performed under the care of the senior author at Waikato Hospital within the study period was utilized. Patients who had undergone comprehensive ESS (i.e., complete sphenoethmoidectomy with frontal recess dissection), regardless of the indication for surgery, were included in the study. There were no exclusion criteria.

The following clinical details were collected: age at the time of surgery, gender, ethnicity, diagnostic subgroup as per the European Position Paper on Rhinosinusitis and Nasal Polyps 2020 [16], procedure details including adjunct surgical procedures, the use of topical and IV TXA, date of discharge, re-presentation to an acute setting within 28 days of discharge, any post-operative imaging done within 28 days of surgery (noting the documented typical time course of post-operative thromboembolic complications [13–15]).

The safety of TXA was measured by assessing rates of thromboembolic complications, specifically pulmonary embolism (PE), deep vein thrombosis (DVT), stroke, and myocardial infarction (MI).

In order to capture outcomes for patients who may have migrated within the region, all patient records were reviewed using Midland Portal, which includes clinical documents from across the Central North Island of New Zealand. Where relevant documents were not available via electronic records, physical records of the patients were accessed.

*2.3. Statistical Analysis*

Descriptive statistics were used to report mean, standard deviation (SD), median, and range. Before statistical analyses were carried out, all patients were de-identified and given a randomly generated study number. All statistical analyses were carried out using SPSS 27 (IBM, Chicago, IL, USA).

**3. Results**

*3.1. Scoping Review*

Once duplicates were removed, the literature search yielded 475 titles and abstracts. Grey literature databases yielded a total of 170 abstracts. A Prisma flow diagram [17] outlining the search results is shown in Figure 1. Three in vitro and two in vivo studies met the eligibility criteria: Dos Reis et al. [18] and Gholizadeh et al. [19] who both used RPMI 2650 human nasal cells, Haghi et al. [20] who used Calu-3 air interface model derived from sub-bronchial human epithelial cell line, Wyrwa et al. [21] who used New Zealand white rabbits, and R Baylis et al. [22] who used Dorset cross sheep. A summary of the available studies is shown in Table 1.

3.1.1. Cytotoxicity

Four studies assessed viability and toxicity of respiratory cells following exposure to TXA. Gholizadeh et al. [19], Dos Reis et al. [18], and Haghi et al. [20] found that both human and animal respiratory epithelia could tolerate a wide range of TXA concentrations ($6.4 \times 10^7$ nM, $2.5 \times 10^7$ nM, and $1 \times 10^5$ nM, respectively). In the in vivo assessments, Wyrwa et al. [21] found that TXA loaded on an electrospun fabric made of Poly ($_L$-lactide-*co*-$_{D/L}$-lactide) had no histologically evident detrimental morphological or cytological effects on the rabbit respiratory epithelium.

3.1.2. Wound Healing Effects

The studies found promising evidence of the wound healing effects of TXA. Haghi et al. [20] found that the amount of wound closure was significantly greater in TXA-treated sub-bronchial human epithelial cells (near 100%) compared to no treatment (63%) following 24 h of wound induction ($p \leq 0.05$). Dos Reis et al. [18] found that adding TXA significantly reduced the wound size in human nasal cells by $90.5 \pm 5.6\%$ of its initial size. Gholizadeh et al. [19] also observed increased wound healing in human nasal cells where an aqueous

form of TXA or chitosan-TXA formulation was used. The results also indicated that the addition of other substances, such as chitosan or hyaluronic acid, may enhance the wound healing features of TXA.

### 3.1.3. Anti-Inflammatory Effects

There were some promising data but no consensus on the anti-inflammatory effects of tranexamic acid. Dos Reis et al. [18] found a significant decrease in oxidative status of the human nasal epithelia cells when treated with TXA at a concentration of 0–40 mg/mL. They also found a significant decrease in IL-8 cytokines, but not IL-6. Similarly, Haghi et al. [20] found a significant decrease in the secretion of IL-1$\beta$, IL-8, and INF-$\gamma$ in sub-bronchial human epithelial cells compared to no treatment.

On the other hand, Baylis et al. [22] assessed biopsies of turbinate injuries in sheep 48 h after exposure to a self-propelling formulation of thrombin and TXA (SPTT). They found no significant difference in pathologist scores with regards to inflammation, ulceration, and infiltration and inflammatory debris, in the exposure or control group containing thrombin only (Floseal). Wyrwa et al. [21] studied three biopsies of the rabbit nasal septum and the overlying mucus membrane. Contrary to all the other studies, they found no evidence of inflammatory processes in the control or the PLA-TXA treated groups. It is not reported whether this finding was an artefact of the study. Overall, there is limited evidence assessing the anti-inflammatory effects of TXA. However, the available evidence implies a potential beneficial effect that merit further investigation.

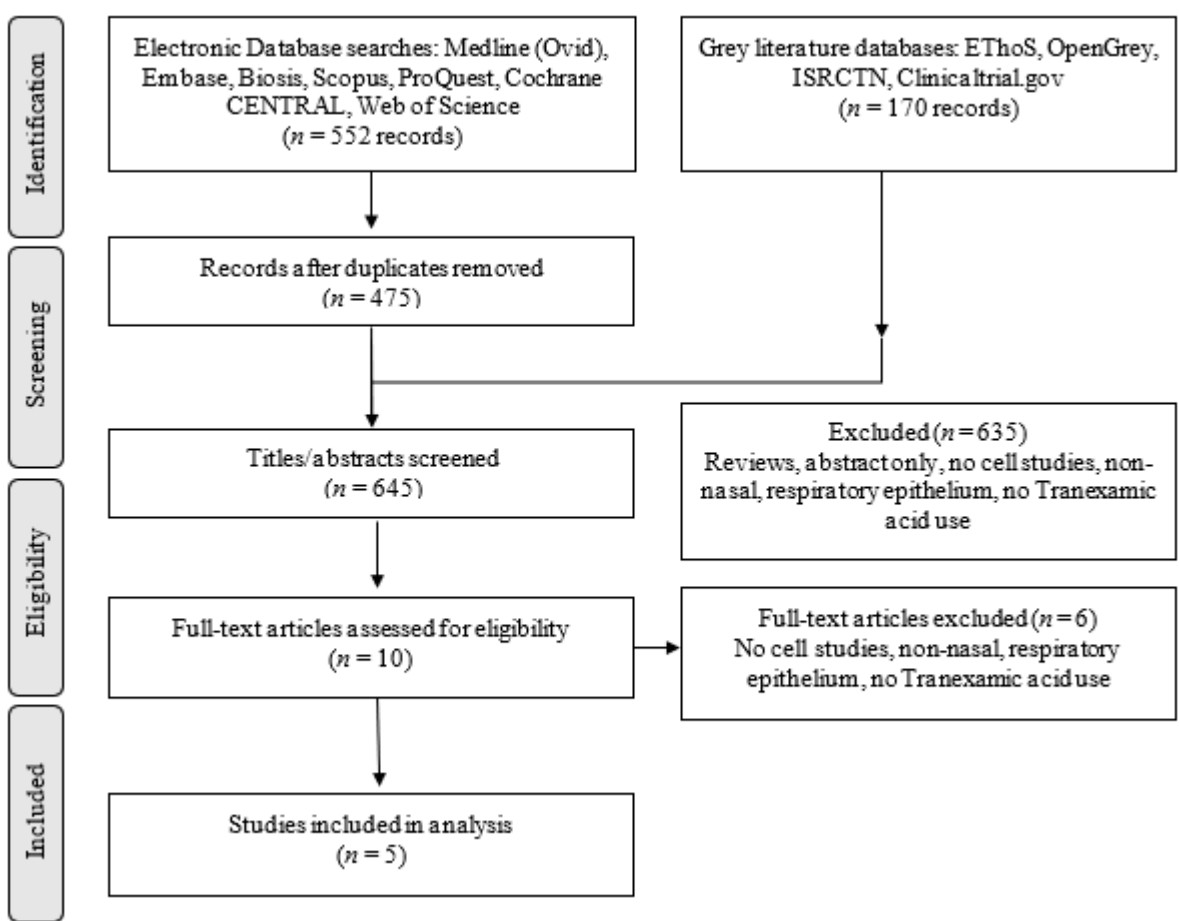

**Figure 1.** Prisma Flow Diagram [17]—scoping review.

**Table 1.** Characteristics of included studies—scoping review.

| Study | Objective | Study Design | Cell Lineage Used | TXA Strength and Formulation | Main Finding |
|---|---|---|---|---|---|
| Dos Reis et al. [18] | Develop a novel nasal powder formulation combining TXA with hyaluronic acid (HA) as a wound healing agent to repair the nasal mucosa. | In vitro | RPMI 2650 nasal cells | 40 mg/mL TXA vs. TXA + 0.1, 0.3 and 0.5% hyaluronic acid (TXA + HA) | No cytotoxicity at 0–40 mg/mL concentrations of TXA. Significant increase in cell proliferation above 40 mg/mL but no cytotoxicity. Use of TXA reduced the wound size to $90.5 \pm 5.6\%$ ($p < 0.05$) of the initial size. Addition of HA further improved healing properties Significant decrease in IL-8 but not IL-6 cytokines. |
| Gholizadeh et al. [19] | Investigate the application of a thermosensitive Chitosan-based formulation loaded with TXA for nasal epithelial wound healing. | In vitro | RPMI 2650 nasal cells | Chitosan (biodegradable polymer) containing TXA (1%) formulation vs. TXA 1% ($w/v$) aqueous solution | 89.32% cells still viable at $1 \times 10^7$ nM concentration of aqueous TXA 55.12% of cells still viable at $1 \times 10^7$ nM concentration of CS-TXA formulation Rate of wound healing in CS+TXA faster but after 6 h showed no statistical difference to aqueous-TXA alone |
| Haghi et al. [20] | Investigate a novel anti hemorrhagic drug delivery technology, containing TXA as a fine aerosol for delivery directly to the lung. | In vitro | Calu-3 air interface model (sub-bronchial human epithelial line) | Aqueous TXA solution (20 mg/mL) spray dried (2% $w/v$) vs. no treatment | Histological examination after 4 weeks of exposure to poly-TXA revealed absence of inflammatory effects and no changes in cellular morphology of the cartilage tissue, mucosal membrane and epithelial cells in the rat. |
| Wyrwa et al. [21] | Evaluate a novel electrospun material based on poly (L-lactide-co-D/L-lactide) (PLA) loaded with hemostatic agents in in vitro and in vivo experiments. | In vivo | New Zealand white rabbits | PLA (electrospun polymer dressing made of Poly) containing TXA 20% ($w/w$) | No significant differences in pathologist scores for inflammation between SPTT sites and control sites. Treatment of turbinate injuries with SPTT did not cause any more local tissue damage than plain gauze D-dimer levels did not differ significantly between any time points or between sheep that received SPTT or Floseal to their carotid injuries ($p > 0.05$). |
| R Baylis et al. [22] | Investigate a self-propelling formulation of thrombin and TXA (SPTT) in stopping bleeding in paranasal sinus injury in sheep. | Interventional animal study | Dorset cross sheep | 0.34 mg human thrombin + protonated 375 μL TXA (SPTT) on a gauze vs. Floseal (dressing containing human thrombin) OR Plain gauze | 82.2% ($\pm$22.13%) of cells still viable at $1 \times 10^5$ nM. Wound closure greater in TXA treated wounds 24 h after wound induction Significant decrease in secretion of IL-1β, IL-8 and INF-γ cytokines where TXA was used. |

## 3.2. Retrospective Clinical Study

### 3.2.1. Patient Characteristics

A total of 720 patients were screened who underwent any surgery under the care of the senior author within the study period. Of these patients, 177 met the eligibility criteria and were included in the final statistical analysis. A patient flow chart is shown in Figure 2.

In the study population, 56.5% (100) were male. 57.6% (102) identified themselves as NZ European/Pākehā compared to 18.1% (32) Māori or NZ European/Māori. Mean age in the study population was 48.8 (±16.7) years.

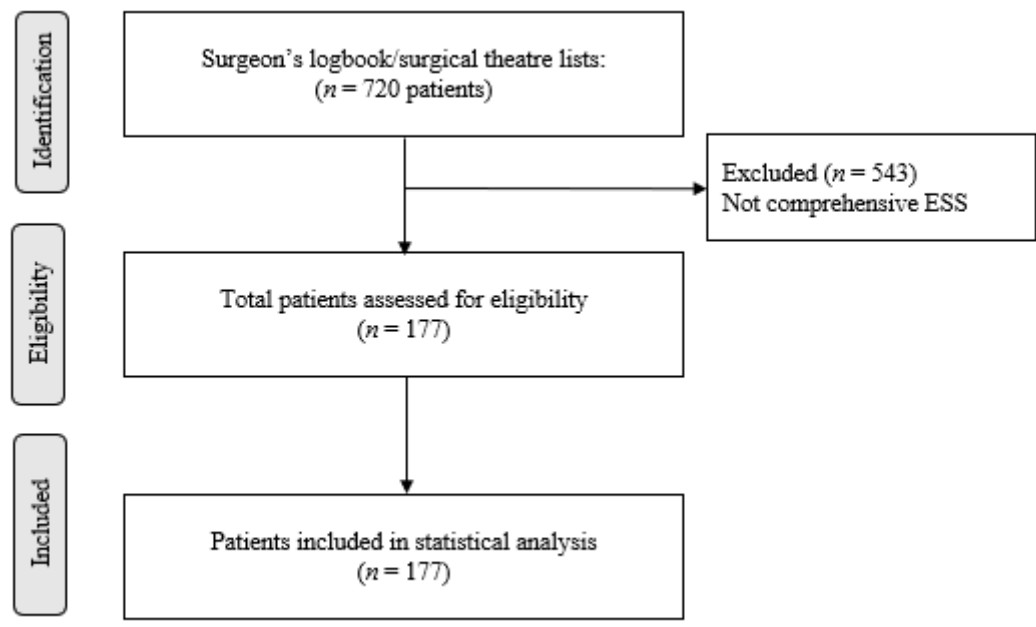

**Figure 2.** Patient flow chart: retrospective clinical study.

3.2.2. Surgical Characteristics

In the study population, 123 (70%) of the patients underwent primary bilateral comprehensive ESS, 36 (20%) revision ESS and 18 (10%) primary unilateral comprehensive ESS. Most patients underwent surgery due to primary diffuse CRS (150, 85%). Clinical details of the study population are shown in Table 2.

**Table 2.** Patient and surgical characteristics: retrospective clinical study.

| | | *n* (%) |
|---|---|---|
| Form of TXA given | Both IV and topical | 159 (89.9) |
| | One form only | 15 (8.5) |
| | Neither form | 3 (1.7) |
| Indication for surgery | Primary diffuse CRS | 150 (84.7) |
| | Primary localized CRS | 3 (1.7) |
| | Secondary diffuse CRS | 7 (4.0) |
| | Secondary localized CRS | 10 (5.6) |
| | Other | 7 (4.0) |
| Type of surgery | Bilateral primary comprehensive ESS | 123 (69.5) |
| | Revision bilateral ESS | 36 (10.3) |
| | Primary unilateral comprehensive ESS | 18 (10.2) |
| Adjunctive surgical procedures | | 112 (63.3) |
| Post-operative imaging | Planned/undergone revision surgery | 5 (2.8) |
| | Under follow-up | 12 (6.8) |
| | Discharged from clinic | 126 (71.2) |
| | Lost to follow-up | 22 (12.4) |

*n*: number of patients or cases; IV: intravenous; TXA: tranexamic acid; CRS: chronic rhinosinusitis.

### 3.2.3. Use of TXA

In our patient population, 159 (90%) patients had received both topical and IV TXA. Only three patients (1.7%) received neither form. Although this was not specifically measurable retrospectively, the avoidance of TXA was primarily due to concerns regarding an increased risk of thrombosis for these individuals.

### 3.2.4. Post-Operative Course

The overall rate of patient re-presentation to an acute setting within 28 days of surgery for all causes was 8.5% (15/177 patients). Of the total complications, the proportion of post-operative bleeding was 20% (3/15 patients).

Where both forms of TXA were given, the rate of re-presentation due to post-operative bleeding was 1.3% (2/159). Over the course of the study period, five (2.8%) patients had planned or underwent revision surgery. All five had received some form of TXA.

### 3.2.5. Safety

There were seven (3.9%) patients who underwent post-operative imaging within 28 days of discharge; three due to respiratory symptoms, one due to neurological symptoms and three others due to routine follow-up for other medical conditions.

In the total study sample, there was only one confirmed case of thromboembolism, in the form of a transient ischemic attack (TIA). This patient had received only topical TXA due to a concern about the risk factors and had acted against clinical advice in stopping his routine anti-coagulation in the post-operative period, prior to the TIA. There were no cases of PE, DVT, or MI identified in the study population over the 28-day post-operative period studied.

## 4. Discussion

Although largely limited to an in vitro context and not tested in human subjects, the scoping review has found no evidence to question the safety of directly applying TXA to respiratory epithelium. It can be concluded, however, that there is a paucity of data in this field, hence why we have proceeded to assess and report our experience. The evidence from the studies cited suggests that respiratory epithelium could tolerate a wide range of TXA concentrations delivered in different formulations, but further study is merited. The clinical practice of the senior author is to use the contents of a single vial of the IV TXA preparation topically, namely 5 mL of 100 mg/mL. This is equivalent to $3.2 \times 10^6$ nM (molar mass of TXA: 157 g/mol), which is within the range of concentrations studied in Gholizadeh et al. [19] and Dos Reis et al. [18]. Furthermore, two recent studies found that the concentration of TXA needed to fully inhibit fibrinolysis in adult human plasma was 14.7 µg/mL and 17.5 µg/mL respectively [23,24]. This is equivalent to 93.5 nM of TXA, an even lower concentration.

TXA exhibited wound healing effects when applied directly to respiratory epithelia in vitro. These findings are consistent with data from studies of other types of epithelia. In ex vivo human keratinocyte studies, complete re-epithelialization was seen after limited exposure to TXA at 100 mg/mL. The healing time was no slower than what was found for saline control group [25]. In skin models where damage was caused by chemical and UV rays, TXA was shown to significantly accelerate barrier recovery [26]. In a rat surgical model, topically applied TXA was shown to have better bone healing capabilities compared to systemic TXA and the control [27].

The potential anti-inflammatory effects of TXA on respiratory epithelia [18,20] are also similar to what is found in the literature for other types of epithelia. In an intestinal epithelial trauma/hemorrhagic shock model, TXA was shown to protect the gut barrier [28]. In a mouse model of rosacea, treatment with TXA significantly attenuated pro-inflammatory cytokines (IL-6, TNF-$\alpha$) and MMP9 expression, thus ameliorating the symptoms [29]. This highlights that potentially the anti-inflammatory effects of TXA could be of significant benefit after ESS, and thus warrants further investigation. It is also noted that the effect

of TXA on olfactory neuroepithelium (specifically ciliary function) and the bone of the nose and sinuses were not addressed in this review and merit further study before more definitive comments can be made about the local safety of topical use in this context.

The clinical data that we present imply that TXA use in ESS is safe with regards to thromboembolic events over a more appropriate follow-up period of 28 days. That said, given that thromboembolic events are rare complications, this study is underpowered to provide absolute assurance of safety. The low overall rate of thromboembolism is however reassuring and is reflected in several meta-analyses assessing combined topical and IV TXA use in knee and hip arthroplasty [30–34]. Although the safety of IV TXA alone in medical and surgical interventions appears to be largely proven [35], there are a paucity of studies, outside of orthopaedic literature, assessing the safety of combined topical and IV TXA, highlighting the need for more research in this area.

To our knowledge, our scoping review was the first to compile an evidence base surrounding the effect of direct application of TXA to respiratory epithelium. While the search was performed by just a single author, we were still able to adequately synthesize the quantitative findings of the available studies.

A strength of the retrospective study is the six-month follow-up period. The retrospective, non-randomized nature of this study means that we were unable to conclude whether addition of topical TXA is superior, equivalent, or inferior to the use of IV only in terms of hemostasis or patient outcomes. We have therefore gone on to complete a prospective randomized controlled trial where we allocated patients to the application of topical TXA or topical normal saline immediately after ESS. All patients received IV TXA. We quantified bleeding, speed of recovery, symptom severity, and endoscopic appearance in both experimental arms [36].

In conclusion, our scoping review reassured us that, based largely on in vitro data, the fragile respiratory epithelium can potentially withstand directly applied TXA, and may in fact benefit from its application with regards to wound healing, but overall we have highlighted the paucity of data in this field. Data from our clinical study support the notion that the combined use of IV and topical TXA is safe with regards to thromboembolic disease. This study highlights the utility of future prospective trials assessing the use of topical TXA at the conclusion of ESS.

**Author Contributions:** Conceptualization, A.R.K. and A.J.W.; methodology, A.R.K. and A.J.W.; formal analysis, A.R.K.; investigation, A.R.K.; resources, A.R.K. and A.J.W.; data curation, A.R.K.; writing—original draft preparation, A.R.K.; writing—review and editing, A.J.W.; supervision, A.J.W.; project administration, A.R.K. and A.J.W.; funding acquisition, A.R.K. and A.J.W. All authors have read and agreed to the published version of the manuscript.

**Funding:** This research was funded by The Waikato Medical Research Foundation, New Zealand (Grant #319).

**Institutional Review Board Statement:** The study was conducted in accordance with the Declaration of Helsinki, and approved by the Auckland Health Research Ethics Committee, protocol code 1242, on 8 May 2020.

**Informed Consent Statement:** Patient consent was waived due to the retrospective nature of the study.

**Data Availability Statement:** All data generated or analyzed during the scoping review are included in this article. Further enquiries can be directed to the corresponding author. The data that support the clinical findings of this study are not publicly available as this was not specifically approved during the Ethics approval process.

**Acknowledgments:** A special thank you for Alana Cavadino from the University of Auckland, School of Population Health for her assistance in interpreting the statistical outputs.

**Conflicts of Interest:** The authors declare no conflict of interest. The funders had no role in the design of the study; in the collection, analyses, or interpretation of data; in the writing of the manuscript; or in the decision to publish the results.

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
