# Peer review of "The Safety of Topical and Intravenous Tranexamic Acid in Endoscopic Sinus Surgery"

_2673-351X, doi:10.3390/sinusitis6020008_

Round 1

Reviewer 1 Report

This is a well written paper. Adhering to PRISMA guidelines. Strength and weaknesses are addressed in discussion. I believe the presentation is of great clinical interest. 

However I would like to se a better Method section for the retrospective clinical trial. If any reader would like to pick up your method of using TXA they are now at loss with neither dosing for iv and local explained. What types of packs where used? Neither is it clear if the TXA packs replaced standard nasal packing and if removed in recovery where there as such no postsurgical packs? A detailed description is warranted. 

Further Demographics of the study material, age, gender, number of previous sinus surgeries is lacking

The power analysis when looking for rare events has not been addressed in discussion. It would be prudent to mention that the numbers although high for a single site study is not sufficient to rule out risks. 

Author Response

Many thanks for your considered review of our study and we are most grateful for the opportunity to improve this manuscript, based on your comments.

We have expanded the final paragraph of the Introduction to better explain the protocol for TXA administration.

We have added the demographic data and apologise that this was not included having featured in a prior version of this manuscript. Unfortunately we do not have the data about the number of prior procedures for those undergoing revision surgery.

We have added a sentence in the Discussion to acknowledge that this is an under-powered study for detecting rare events.

Reviewer 2 Report

This is a well written manuscript on a very relevant topic. Just a few items of clarification are needed. 

1.     The manuscript does not refer to the approval of the study by the local ethics committee.

2.     Clarify we did the authors use prisma flow diagram, usually used in metaanalisys and systemic reviews.

3.     The authors comment in the discussion that “the scoping review demonstrated that TXA is safe to be applied directly in the sinuses and nasal cavity”. From the scopis review I can  infer that the use of tranexamic acid is safe in vitro. I think it is hasty to say that it is safe to use in sinus and nasal cavity. To make this argument, a human safety study should be conducted.

4.     There is an heterogeneity of the studies included in the scopis review that would mislead conclusions. Please comment on the limitations of this scopis review.

Author Response

Many thanks for your considered review of our study and we are most grateful for the opportunity to improve this manuscript, based on your comments.

The details of the Ethics review process were initially shown in the Methods section but in accordance with the Journals requirements, this had been moved to the "Institutional Review Board Statement" section at the end of the submission where it can be reviewed.

Thanks for commenting of the use of the PRISMA flow diagram. We have presented a Scoping review rather than a Systematic review but the broad principles are the same and so it seemed appropriate to report using this structure. Given this and given that another Reviewer has noted favourably our use of it, we propose to leave this in place. We would be happy to discuss further if you feel strongly that this is not appropriate.

Thank you also for highlighting our slightly lax description of the scoping review and its limitations. We have added comments highlighting that we have found a paucity of relevant data and other comments in the Discussion to acknowledge that we can only make speculative inferences rather than strong conclusions from both the scoping review and the clinical component of this study.

Reviewer 3 Report

  • Great value in addressing common issue of medications used in clinical practice which have limited safety and efficacy data supporting use. 
  • Introduction - how often are patients actually re-presenting? Suggest brief description of TXA mechanism of action
  • Methods needs more detail with respect to the scoping review and statistical analysis in particular.
  • 2.1.2 - perhaps mention what study types were included (only RCTs or otherwise?) Was the screening process performed by one individual or multiple? Any critical appraisal involved in the scoping review? Exclusion criteria does not mention coagulopathies, bleeding disorders, or relevant anti-coagulant medications, possible confounders. 
  • 2.2 - Rationale for 28 days follow-up period? Please explicitly state 
  • 2.3 - was de-indentification and randomisation blinded? Please state. What statistical tests were used to calculate p value? 
  • Discussion - Further weakness includes lack of control for treatments/conditions affecting coagulation.

Please see specific comments below:

Line 22 - re-presented

Line 34 - results 

Line 52 - local and direct mean the same thing?

Paragraph starting at line 55 - perhaps a concluding sentence to make your point that current studies do not have adequate follow-up given the evidence listed prior 

Line 91 - do you mean included other epithelia in addition to respiratory? What is the rationale if evidence for respiratory epithelium is available 

Line 102 - need comma after [16]

3.1 - Figure 1 excludes reviews and abstracts. The exclusions of these article types needs to be explicit in methods 

Line 126 - bolden Figure 1 

Line 292 - maybe just clarify at what time point (ie. after 28 days post op) 

Line 295 - consider rewording to say it is safe to be applied to nasal epithelium

Line 298 - how was this dose decided upon?

Line 313 - what are the anti-inflammatory effects you are referring to, as the results suggest there were no clear findings. Highlight which specifically correlate. 

Line 317 - “this highlights that the potential that” ?

Line 325 - single reviewer should be highlighted in methods. This is not the only difference between a scoping and systematic review. If this paragraph is trying to suggest that a single reviewer scoping review method may be a weakness I think this needs to be reworked.

Line 330 - Consider - “A strength of the retrospective study is the six month follow-up period” 

Author Response

Many thanks for your considered review of our study and we are most grateful for the opportunity to improve this manuscript, based on your comments.

The two requested additions to the Introduction have been made.

2.1.2 We have added that the search was performed by a single individual, the first author. There were no exclusions based on study type or the other possible confounders listed in the review. Your later comment regarding Figure 1 has correctly highlighted however that we did exclude articles where abstracts only were presented and we have adjusted the Methods section to address this omission. Thank you for picking this up.

2.2 The rationale for the 28-day follow-up period is noted.

2.3 Given that this was a retrospective chart review conducted by the first author it was not possible for her to be blinded to the clinical details. Apologies for the incorrect description of the statistical analysis which we have amended. In the text we only present descriptive statistics (mean, median, standard deviation, %) and no inferential statistics (p numbers / statistical tests) were used or presented.

Discussion - we feel that the fact that we were unable to control for e.g. coagulopathies is implicit in the acknowledgment that this is a retrospective study and this was indeed controlled for in our follow-up prospective study which we have now cited.

Specific comments (and thanks also for these corrections):

Line 22 - “re-presented" corrected

Line 34 - “results" corrected

Line 52 - “local and" removed

Paragraph from Line 55 - We feel that this point has already been made earlier in the paragraph with the comment that "Studies that assessed the use of either IV or topical TXA in ESS had a maximum follow-up period of 3-days which was inadequate to sufficiently assess post-operative adverse events"

Line 91 - “non-respiratory epithelium". Apologies for the slightly clumsy wording - this was intended to mean where other tissue types were the exclusive subject of study and has been refined

Line 102 - Comma added after [16]

Line 126 - You have asked us to bolden Figure 1 in the text. We have not seen this as a journal requirement in the instructions for authors; apologies if we missed this. If we are to do this then presumably it makes sense to do the same for Table 1 / Table 2 / Figure 2 as well or not at all? For the moment we have not actioned this

Line 292 - comment added

Line 295 - re-worded and expanded

Line 298 - rationale explained

Line 313 - toned down and referenced

Line 317 - corrected

Line 325 - this comment probably does not add to the article and so has been removed / re-written

Line 330 - changed, thank you

Reviewer 4 Report

This interesting and important submission is reported to be a scoping review coupled with a retrospective study to investigate the safety and potential benefits of tranexamic acid (TXA) given via both IV and topical routes during endoscopic sinus surgery. The scoping review is designed “to assess the effects of TXA when applied directly to respiratory epithelia.” The retrospective study is meant “to analyze the safety of using both IV and topical TXA in ESS.” The authors conclude that the use of IV and topical TXA is safe and may even have positive effects on wound healing and inflammation.

Comments

1.     The authors state that they began with a scoping review of the existing literature on the safety of topical TXA in regards to the respiratory epithelia. However, it is not clear whether “scoping review” is the correct terminology for their study, or at least perhaps their justification for calling it a scoping review does not seem correct. They state “With the search performed by a single author, this review is termed a scoping review rather than a systematic review.” It is unclear whether this characteristic is what defines a scoping review. As Munn et al report in their publication on the nomenclature, with well over 3,000 citations, “Researchers may conduct scoping reviews instead of systematic reviews where the purpose of the review is to identify knowledge gaps, scope a body of literature, clarify concepts or to investigate research conduct. While useful in their own right, scoping reviews may also be helpful precursors to systematic reviews and can be used to confirm the relevance of inclusion criteria and potential questions.” While that would seem to be the correct purpose for a review in this particular clinical area – with its relative dearth of information – the authors do not seem to be using this article to necessarily identify gaps in the literature, but rather to make fairly strong (and arguably unsubstantiated) claims about the efficacy and safety of topical TXA when used during sinus surgery when in fact that evidence does not clearly exist, except in a few in vitro/vivo studies. Coupling this with an uncontrolled retrospective study to make this claims is somewhat misleading as they seem to imply more robust findings than we can prove with their data. At best, a human nasal cell line study was evaluated in the review, while the other studies utilized bronchial cells, sheep, and rabbit specimens. Therefore, the conclusions that have been drawn may be overly generous and should be tempered significantly, with an emphasis on the lack of in vivo human data.

a.     Munn et al. Systematic review or scoping review? Guidance for authors when choosing between a systematic or scoping review approach. BMC Medical Research Methodology volume 18, Article number: 143 (2018). https://bmcmedresmethodol.biomedcentral.com/articles/10.1186/s12874-018-0611-x

2.     As far as the safety of IV TXA in regards to the risk of thrombosis, the authors are correct that this has been well studied in the past. The additional question of the safety of topical TXA used in addition to the IV route is an important one, but is not necessarily addressed robustly in either the “scoping” review nor in the retrospective study. As discussed above, the non-human and in-vitro/vivo studies do not necessarily demonstrate an absence of safety concerns when used in vivo in a human patient.

3.     Furthermore, in regards to the potential risk of damage to epithelium, has any evaluation been done to any potential damage to the olfactory neuroepithelium? This is a different cell type than nasal respiratory epithelium, and did not seem to be included in the scoping review. What about any effect on the function of the cilia themselves? The cited in vitro/vivo studies mostly focus on wound healing and size of wound when supporting the efficacy of TXA but not necessarily the functional viability or function of the tissue.

4.     The senior author uses IV TXA at the beginning of the case and then topical TXA applied via cotton pledgets at the end. It is not clear whether this topical formulation is similar to or different to the ones reported in the “scoping” review, in which TXA powder was utilized by Dos Reis et al – and also was used in combination with hyaluronic acid in that study. If the formulation and method of delivery in the retrospective study is indeed different, and the review studies used TXA in combination with another compound (i.e. hyaluronic acid), then it isn’t entirely clear that we can extrapolate their findings of tissue safety to the present study, as we do not know to which substance we should attribute those findings.

5.     Was “TXA” included as a search term in the review query, along with other abbreviations such as AMCA and t-AMCA? If not, why not?

6.     A very important and well-cited review to support the safety of IV TXA would be worth including:

a.     Taeuber et al. Association of Intravenous Tranexamic Acid With Thromboembolic Events and Mortality. A Systematic Review, Meta-analysis, and Meta-regression.  JAMA Surg. 2021;156(6):e210884. https://jamanetwork.com/journals/jamasurgery/fullarticle/2778639

7.     In the retrospective portion of the study, 90% of included patients received topical TXA in addition to IV TXA. Given the vast majority of patients in this study received both, it is difficult to draw any conclusions about the relative safety of topical TXA, because we have no robust control group with which to compare it. Were any statistical comparisons possible between those who received topical TXA and those who did not? All 5 patients who were eligible for revision surgery had received topical TXA. We don’t know whether that is simply due to the sheer numbers of those who received topical TXA which far outweighed those who did not, or whether topical TXA predisposed them to some sort of functional deficit that required revision surgery. Presumably the former is more accurate, but this limitation should be acknowledged.

8.     Overall, I would argue that the statements and tone in the manuscript should be more measured given the relatively lack of robust evidence. I agree with the authors’ final conclusions that further prospective randomized studies are warranted. But we need to be more careful before we make such strong claims that topical TXA is safe, reduces inflammation, and improves outcomes. For example, as above, do we have any data for olfactory outcomes? If it turns out topical TXA has a similar effect as Zicam on the olfactory epithelium, we would be doing a disservice to many of our patients. To be clear, while I don’t believe this is actually the case based on its mechanism, I also see no data in this manuscript that would tell me otherwise, and I would simply caution the authors to be a bit more balanced in how they report their findings. We also need to acknowledge that the scoping review included only in vitro or non-human in vivo studies.

Author Response

Many thanks for your considered review of our study and we are most grateful for the opportunity to improve this manuscript, based on your comments.

Thank you for your comments on nomenclature regarding the scoping review. On reflection we are not sure that the sentence in question in the Discussion adds to the article and so have therefore removed it.

The comments around the certainty attached to any conclusions are entirely reasonable and align with the other Reviewers comments around the ability to draw conclusions from both aspects of this article. We have made a number of changes to the Discussion section to hopefully address these concerns. We hope that this now represents a more appropriate representation of what can reasonably be concluded from our study

We acknowledge that there was no specific study of the the effect of topical TXA on olfactory epithelium and this is now noted in the Discussion section also.

Point #4 - we have expanded the description of the technique used for topical TXA application in the Introduction section.

Point #5 - you are correct in highlighting that “TXA” was not used as a search term and in hindsight this was perhaps illogical given the other abbreviations included. We speculate that it is unlikely that this caused any relevant studies to be missed but have no explanation other that this was potentially an oversight

Point #6 - we have referenced this article - thanks

Point #7 - we have not drawn any conclusions from this observation which as you state may well merely reflect the small sample size and non-randomised nature of this study. We have added a comment in the Discussion that the use of topical TXA may in fact generate inferior outcomes but given the above issues with the observation regarding the 5 revision cases it seems inappropriate to elaborate on this further.

Point #8 - hopefully this has been adequately addressed with the above comments and associated changes to the manuscript

Round 2

Reviewer 4 Report

This interesting and important submission is reported to be a scoping review coupled with a retrospective study to investigate the safety and potential benefits of tranexamic acid (TXA) given via both IV and topical routes during endoscopic sinus surgery. The scoping review is designed “to assess the effects of TXA when applied directly to respiratory epithelia.” The retrospective study is meant “to analyze the safety of using both IV and topical TXA in ESS.” The authors conclude that the use of IV and topical TXA is likely safe and may even have positive effects on wound healing and inflammation. I appreciate their incorporation of the recommendations previously suggested. I do have some further feedback.

Comments

1.     In the introduction, the authors state “Given the relative abundance of data regarding the efficacy of TXA in ESS but the paucity of appropriate safety data, the primary objective of this study was to evaluate the safety of TXA in ESS, both with regards to local effects on the sinonasal mucosa and thromboembolic complications.” There is actually a systematic review and meta-analysis that includes both topical and IV TXA and looks at both efficacy and safety in ESS that they may wish to reference. The included studies do not look at the local effects on sinonasal mucosa which distinguishes this present study (although, as previously noted, the scoping review also does not identify anything studies using in vivo human subjects, so this claim remains to be clinically proven).

a.     Pundir et al. Role of tranexamic acid in endoscopic sinus surgery - A systematic review and meta-analysis. Rhinology 51: 291-297, 2013.

2.     The authors acknowledge that they did not use “TXA” as a search term despite using other abbreviations such as AMCA. I would encourage the authors to re-run the database query to determine whether additional studies are identified.

3.     In the patient characteristics section of the retrospective study, the authors state “Most frequent age to undergo surgery was 58 years.” This would be the mode. I assume they want to say “The mean age to undergo surgery was 58 years.” Please clarify.

4.     In the discussion, the authors state “Although largely limited to an in vitro context and not tested in human subjects, the scoping review implies that TXA is likely to be safe to apply directly to respiratory epithelium. It can be concluded however that there is a paucity of data in this field, hence why he have proceeded to assess and report our experience.” The retrospective review does not in fact provide any evidence to support the safety of applying TXA directly to the respiratory epithelium in terms of local tissue health. The tissue of the patients in the study was not evaluated to support this.

5.     They then state “The evidence from both human and animal studies suggest that respiratory epithelium could tolerate a wide range of TXA concentrations delivered as different formulations.” There are no human studies to support this, only human in vitro cell line assays.

6.     I appreciate that the authors have acknowledged that the scoping study did not evaluate the effect of TXA on olfactory tissue, but I think it is also worth noting that there has never been any mention of the effect of TXA on respiratory epithelium ciliary function – only wound healing. This provides no indication as to the effect on the actual functionality of the tissue. It may be worth simply adding a limitations section at the end of the discussion as most manuscripts contain – it is notable that this manuscript lacks such a section. The follow-on randomized controlled trial that the authors mention at the end of the discussion would seem to potentially address some of these concerns, but that does not eliminate the limitations of the present study.

7.     The citation for the authors’ follow-on RCT is listed as reference 35 in the manuscript, but it should in fact be reference 36.

Author Response

Thanks for the further review of our manuscript which has helped us to improve the manuscript further. We hope that this will now be considered worthy of publication. With regards to the specific comments:

  1. Thanks for highlighting this. We were aware of that article but felt it more appropriate to reference the more recent meta-analysis and systematic review of systematic tranexamic acid use (reference #6) and to critically examine the limitations of the topical tranexamic acid literature specifically (references #7-10). The 2013 Pundir et al article merely offers two sentences regarding safety: “Pooled data for post operative nausea or vomiting showed 6 patients out of 56 suffered with nausea and vomiting in tranexamic acid group compared to 4 out of 56 patients in the control group (32,36). None of the studies reported serious thromboembolic event in any patients both in tranexamic acid and controls.” 
  2. Thanks again for highlighting this. My student (first author) has now finished her Honours year and left the University so does not have direct access to the software to do further searching. I can however reassure you that the search that was done searched 64 different terms that Ovid records as being other terms used for tranexamic acid. We have attached with the re-submission the details of this search.
  3. 58 years was indeed the mode. Given that this has caused confusion and adds little to the manuscript however we have opted to remove this.
  4. Thanks again for your assistance with ensuring that our conclusions are robust and evidence-based. We have adjusted the first sentence of the Conclusion in line with your comments.
  5. As per the comments immediately above we have also edited the third sentence of the Conclusion in line with your comments.
  6. Thanks for reinforcing this point. We have added further comments in the 3rd paragraph of the Conclusion, also highlighting other tissue types within the nose and sinuses that have not been studied.
  7. Thank you for identifying and correcting this error.
